# AV-GS: Learning Material and Geometry Aware Priors for Novel View Acoustic Synthesis

**Swapnil Bhosale**
University of Surrey, UK

**Haosen Yang**
University of Surrey, UK

**Diptesh Kanojia**
University of Surrey, UK

**Jiankang Deng**
Imperial College London, UK

**Xiatian Zhu**
University of Surrey, UK

## Abstract

Novel view acoustic synthesis (NVAS) aims to render binaural audio at any target viewpoint, given a mono audio emitted by a sound source at a 3D scene. Existing methods have proposed NeRF-based implicit models to exploit visual cues as a condition for synthesizing binaural audio. However, in addition to low efficiency originating from heavy NeRF rendering, these methods all have a limited ability of characterizing the entire scene environment such as room geometry, material properties, and the spatial relation between the listener and sound source. To address these issues, we propose a novel *Audio-Visual Gaussian Splatting* (AV-GS) model. To obtain an implicit material-aware and geometry-aware condition for audio synthesis, we learn an explicit point-based scene representation with an audio-guidance parameter on locally initialized Gaussian points, taking into account the space relation from the listener and sound source. To make the visual scene model audio adaptive, we propose a point densification and pruning strategy to optimally distribute the Gaussian points, with the per-point contribution in sound propagation (e.g., more points needed for texture-less wall surfaces as they affect sound path diversion). Extensive experiments validate the superiority of our AV-GS over existing alternatives on the real-world RWAS and simulation-based SoundSpaces datasets. Project page: `https://surrey-uplab.github.io/research/avgs/`

## 1 Introduction

Novel view synthesis [25, 1, 14] allows to generate images for any target viewpoints, which has been extensively studied and advanced. For real-world applications in augmented reality (AR) and virtual reality (VR), solely visual rendering of 3D scenes without spatial audio (i.e., deaf) fails to fully immerse users in the virtual environment. This thus inspires a recent surge of investigating novel view acoustic synthesis (NVAS) [4, 17, 5, 32]. By synthesizing binaural audio (two channels corresponding to the left and right ear) taking into account the factors like directionality, distance and relative elevation, this can create a spatial audio experience akin to real-life perception [23]. However, realistic binaural audio synthesis is challenging, since the wavelengths of sound waves are much longer, necessitating the modeling of wave diffraction and scattering. To illustrate, while blocking the sun with your thumb is easy, blocking thunder sounds with your thumb is difficult because sound waves wrap around obstacles. A sound wave propagating through a 3D space undergoes various sophisticated acoustic phenomena, like direct sound, early reflections, and late reverberations.

Aiming to render binaural audio from the mono audio for target poses, Neural Acoustic Field (NAF) [21] learns room acoustics in a synthetic environment with a 2D grid of implicit representations as a condition. However, deriving a 2D representation grid for real-view scenes is extremely challenging in the presence of unconstrained objects, materials, and occlusion in 3D scenes. Alternatively, AV-NeRF

38th Conference on Neural Information Processing Systems (NeurIPS 2024).

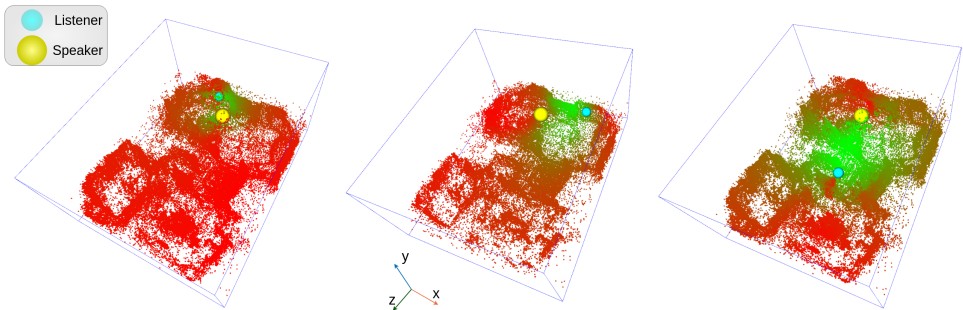

Figure 1: Sound propagation point patterns between a listener (blue sphere) and emitter (yellow sphere) captured by our AV-GS. Notice the points outside the propagation path (points behind the speaker, points behind rigid walls). Please note we slice the scene into half along the y-axis (omitting the points from the ceiling) in order to facilitate better visibility.

[17] leverages implicit representations of the visual cue by learning a vision NeRF model [25] that generates the listener's view. Nonetheless, this can only offer limited conditions, as the listener's view provides information from only the line-of-sight, whilst audio spreads around corners more widely (a listener cannot see behind but can hear sources that are behind). Further, NeRF's training and inference efficiency is typically low [37, 14].

To overcome these limitations, in this work, we present a novel *Audio-Visual Gaussian Splatting* (AV-GS) model, characterized by efficiently learning an explicit, holistic 3D scene condition with rich material and geometry information, as well as more comprehensive contextual knowledge beyond the listener's field of view for enhanced synthesis conditioning. Due to the intrinsic discrepancy between vision and audio as discussed earlier, extending existing 3D Gaussian splatting-based (3DGS) [14] from visual scenes to 3D audio (*i.e.*, spatial audio) is non-trivial. Optimizing towards scene visual reconstruction, points tend to over-populate along object edges and under-populate in texture-less regions such as walls, doors etc. But, such point distribution is rhetoric when learning sound propagation since, major changes in sound paths (absorption and diversion) primarily happen around those texture-less regions. The key challenge lies in jointly modeling 3D geometry and implicit material characteristics of the visual scene objects to instigate direction and distance awareness for realistic binaural audios. To that end, we decouple the physical geometric prior from the 3DGS representation to learn an acoustic field network by introducing audio-guidance parameters. These audio-guidance parameters combined with their relative distance and direction from the listener and sound source, are projected on-the-fly to derive holistic scene-aware and implicit material-aware conditions for synthesizing binaural audios (see Figure 1). We hypothesize that converging AV-GS on binaural audio reconstruction loss, requires location and density adjustment of the Gaussian points relative to the "audio-guidance" provided by the individual point. Towards this direction, we design a Gaussian point densification and pruning strategy based on the individual per-point contribution in providing this "guidance" for sound propagation, ultimately improving the overall binaural audio synthesis.

To summarize, we make the following *contributions*: (1) The first novel view acoustic synthesis work with conditions on holistic scene geometry and implicit material information; (2) A novel AV-GS model that learns a holistic geometry-aware material-aware scene representation in a tailored 3DGS pipeline; (3) Extensive evaluations validating the advantages of our method over prior art alternatives on both synthetic and real-world datasets.

## 2 Related Work

**Audio binauralization** converts monaural (single-channel) audio signals into binaural (two-channel) audio signals. It can enhance immersion by simulating sound directionality, distance, and spatial cues. Generating realistic binaural audio from mono audio is challenging [31, 30], prompting the exploration of various conditioning techniques in the literature.

*Geometry and material conditioning* Luo et al. [21] introduced Neural Acoustic Field (NAF) for room acoustics modeling using implicit neural representations of geometric features, capturing spatial information of speakers and receivers. Su et al. [34] also learned implicit neural representations for audio, focusing on interactive acoustic radiance transfer, relying on scene geometry as input. Anton and Dinesh [29] proposed a material-aware binaural sound propagation model, incorporating

material and topology data, necessitating an acoustic material database. In contrast, our approach learns explicit representation from 3D scenes and disentangles learning of physical and (implicit) material properties, eliminating reliance on scene geometry and material characteristics as inputs.

*Visual cue conditioning* Recent works leverage visual cues to generate spatial audio from mono audio, capturing complementary scene characteristics [9, 38, 27, 41, 18, 43, 19, 28]. Li et al. [15] proposed a multi-task approach optimizing binaural audio generation and flipped audio classification, sharing visual cues from corresponding video frames. Chen et al. [4] synthesized sound by analyzing input audio-visual cues, incorporating active speaker features, target pose, and encoded visual features in the acoustic synthesis pipeline. Alternatively, Liang et al. [17] encoded the listener's position and orientation, conditioned by the listener's view. It additionally provides local visual depth information dependent on the listener's position. However, focusing solely on the listener's view overlooks the broader 3D scene geometry's contribution, which is crucial for sound propagation. We thus propose to learn a holistic 3D scene representation, enhancing binauralization guidance with additional audio parameters.

**3D Scene Representation Learning** Point-based rendering techniques, initiated by [10], utilize point-based representation where each point affects a single pixel. Zwicker et al. [45] advanced this with ellipsoid-based rendering (splatting), allowing mutual overlap to fill image holes. At the absence of given geometry, Mildenhall et al. [25] explored neural implicit representation, NeRF, predicting view-dependent radiance via implicit density fields. NeRF requires combining colors of densely sampled points along the camera rays for high-quality rendering. 3D Gaussian splatting (3D-GS) [14], a novel-view synthesis method, employs explicit point-based representation, contrasting with NeRF's volumetric rendering. Since its real-time high-quality rendering capabilities that 3DGS has been applied to various domains, including simultaneous localization [13, 24], content generation [36], and 4D dynamic scenes [16, 39, 42], among others.

However, no works take the 3DGS advantages to improve the NVAS task. Hence, in this work, we for the first time exploit the 3D-GS for NVAS, to the best of our knowledge, including capturing the scene geometry and material-related information for audio signal processing purposes. Further, we adapt the point management mechanism to account for the characteristics of sound propagation. In a nutshell, our approach aims to improve binaural audio reconstruction loss in AV-GS by adjusting Gaussian point location and density based on their contribution to sound propagation guidance.

## 3 Method

**Problem definition** Deployed at a location $X_S$ in a 3D scene, a sound source $S$ emits a mono audio $a_{mono}$. A listener entity $L$ moves around in this 3D scene, capturing multiple observations using a camera mounted with a binaural microphone. Each observation $O_p = (V_C, V_A)$, constitutes a pair of a camera view $V_C$ and an auditory perspective $V_A$, w.r.t a specific listener pose $p = (X_L, d)$, where $X_L$ is the 3D position of the listener and $d$ is the viewing direction corresponding to listener's head. A camera view $V_C$ defines that at the pose $p$, the listener would observe a RGB image, $I$ (as their view). Similarly, an auditory perspective $V_A$ defines that at the pose $p$, the listener hears a binaural audio, $a_{bi} = (a_l, a_r)$ where $l$ and $r$ represent the left and right ear respectively. Given $N$ observations from the listener, $O = \{O_1, O_2, ..., O_N\}$, the task is to predict the binaural audio $a_{bi}{}^*$ for a novel observation $O_{p^*}$ having an arbitrary unseen pose $p^*$.

### 3.1 Audio-Visual Gaussian Splatting (AV-GS)

Our model is comprised of a 3D Gaussian Splatting model $G$, an acoustic field network $\mathcal{F}$, and an audio binauralizer $\mathcal{B}$. In order to model sound propagation in space and time, having a holistic scene prior as contextual guidance is essential. Crucially, this guidance must incorporate both geometric and material-related characteristics. For facilitating the learning of visual and auditory modalities with inherently distinct characteristics, we decouple the physical geometry and the acoustic field by introducing in-between an *acoustic field network* $\mathcal{F}$ that learns geometry-aware and implicit material-aware scene contexts.

In the following, we first, provide a brief regarding learning the scene geometry. Later, we explain how to construct an audio-focused representation $G_a$. We further describe the process of decoding the parameters of $G_a$ on the fly using view-dependent information, followed by point densification and pruning tailored for sound propagation.

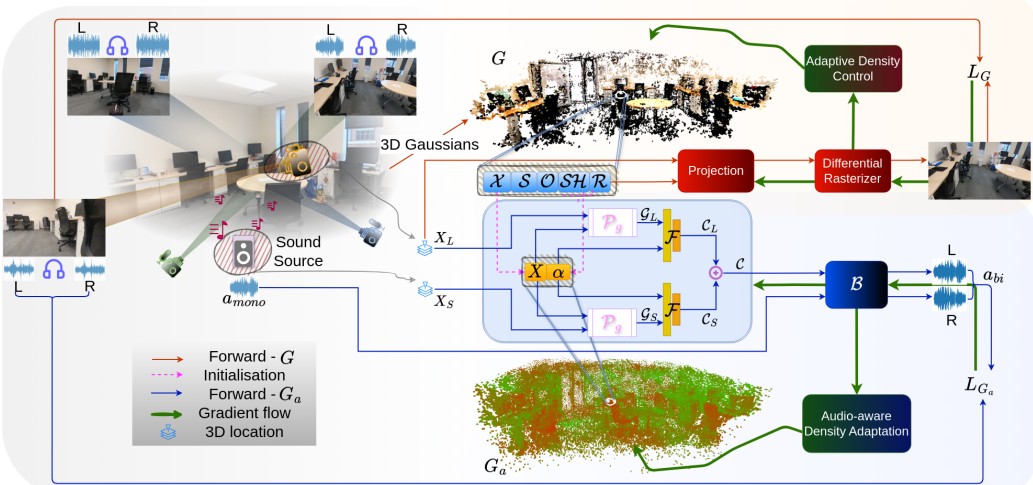

Figure 2: Overview of our proposed AV-GS. Our model is comprised of a 3D Gaussian Splatting model $G$, an acoustic field network $\mathcal{F}$ and an audio binauralizer $\mathcal{B}$. We first train $G$ to capture the scene geometry information. Next, we construct an audio-focused point representation $G_a$, with the location $X$ and audio-guidance parameter $\alpha$ initialized by the pre-trained $G$. Then the acoustic field network $\mathcal{F}$ is used to process the $\alpha$ parameters for all the Gaussian points in the vicinity of the listener and the sound source (in the 3D space). The output from $\mathcal{F}$ is finally used to condition the audio binauralizer $\mathcal{B}$, which transforms the mono audio to binaural audio w.r.t the listener and sound source location.

**3D Gaussian Splatting** (3D-GS) [14] is a state-of-the-art novel-view synthesis method that learns an explicit point-based representation of the 3D scene. It is utilized here to capture the scene geometry. Specifically, each point in the explicit representation $G$ is represented as a 3D Gaussian ellipsoid to which physical attributes like position $\mathcal{X}$, quaternion $\mathcal{R}$, scale $\mathcal{S}$, opacity $\mathcal{O}$ and Spherical Harmonic coefficients ($\mathcal{SH}$) representing view-dependent color are attached. For an arbitrary camera view $V_C{}^*$ with a pose $p^*$, we project/splat 3D Gaussians onto the 2D image plane to obtain the listener's view as an RGB image $I^*$. The projection of 3D Gaussian ellipsoids can be formulated as:

$$\Sigma' = JW\Sigma W^T J^T \tag{1}$$

where $\Sigma'$ and $\Sigma = RSS^T R^T$ are the covariance matrices for 3D Gaussian ellipsoids and projected Gaussian ellipsoids on 2D image from a viewpoint with viewing transformation matrix $W$. $J$ is the Jacobian matrix for the projective transformation. Please refer to [14] for more details on splatting.

### 3.1.1 Acoustic Field Network

As a part of the decoupling approach, we further introduce an audio-focused point-based representation $G_a$. Concretely, every point in $G_a$ is parameterized using the location $X$, alongside a learnable audio-guidance parameter $\alpha$ to encapsulate implicit material-specific characteristics of the scene. $X$ is initialized using the location of Gaussian points in $G$, in order to impose the scene geometry knowledge. To initialize $\alpha$ we concatenate the view-dependent color priors from $G$, particularly spherical harmonics $\mathcal{SH}$ and quaternion feature $\mathcal{R}$. An intuition is to choose the parameters that provide information regarding the color and density characteristics, which are often correlated with material properties [8] (see ablation on the choice of different parameters from $G$ for forming $\alpha$ in Section 4.4).

In order to drive the mono to binaural audio transformation, a pose-specific holistic scene context has to be derived from $G_a$. To achieve this, we derive a position-guidance feature $\mathcal{G}$ w.r.t both, the listener $L$ and the sound source $S$. Position-guidance $\mathcal{G}$ concatenated with the audio-guidance $\alpha$ is used as an input to the acoustic field network $\mathcal{F}$ to obtain a joint context for each point w.r.t to the listener and the sound source separately.

$$\mathcal{C} = \mathcal{C}_S \oplus \mathcal{C}_L \tag{2}$$

$$\mathcal{C}_i = \mathcal{F}(\alpha, \mathcal{G}_i); \mathcal{G}_i = \frac{X - X_i}{\|X - X_i\|_2}, i \in \{S, L\} \tag{3}$$

Concatenating (represented using $\oplus$) the listener-context and the speaker-context yields the overall pose-specific holistic scene context. We obtain the condition for binauralization by averaging the context across all points in $G_a$, post dropping points outside the vicinity of the listener and sound source. The vicinity for the listener and sound source is defined by the nearest $k_L$ and $k_S$ points based on the Euclidean distance between $X$ and $X_L$ and that between $X$ and $X_S$, respectively. We provide an ablation for selecting the size of vicinity points in Section 4.4.

### 3.1.2 Audio Binauralizer

An audio binauralization module $\mathcal{B}$ transforms the mono audio $a_{mono}$ emergent from the sound source $S$ into binaural audio $a_{bi} = \{a_l, a_r\}$, representing the left and right audio channels. The position and orientation of the listener (relative to the sound source) along with the learned holistic scene context $\mathcal{C}$ is used to condition this transformation.

$$a_{bi} = \mathcal{B}(a_{mono}|\mathcal{C}, X_L) \tag{4}$$

where $X_L$ denotes the listener's position and orientation. We adopt a similar architecture for the binauralization as [17], wherein $X_L$ and $\mathcal{C}$ are fused to generate acoustics masks $m_m$ and $m_d$ representing the mixture and difference of the sound fields respectively. Specifically, an MLP combines $X_L$ and $G_a$ with a frequency query $f \in [0, F]$ to produce a feature vector which is further projected using a linear projection to obtain a mixture mask $m_m$ for frequency $f$. This feature vector is concatenated with the transformed direction $\theta$ and passed to a second MLP to generate a difference mask $m_d$. All frequencies within $[0, F]$ are queried to generate complete masks for both the mixture and difference. The magnitude spectrogram $s_{mono}$, computed using short-time Fourier transform (STFT) over $a_{mono}$, is multiplied with $m_m$ and $m_d$ to obtain the mixture magnitude $s_m$ and the difference magnitude $s_d$, respectively. Finally, an inverse STFT is operated on $s_m$ and $s_d$ to obtain the binaural audios for the left and right channels, $a_l$ and $a_r$. The architecture of the binauralizer is discussed in our appendix A.2.

## 3.2 Model training

Due to our decoupling design, we adopt a dual-stage optimization, starting with an initial warm-up stage that learns the physical properties of the Gaussian points using gradients obtained while reconstructing camera views (RGB images). The objective of this initial stage is to infer an explicit point-based representation $G$ to capture the scene geometry. Optimization of $G$ is adapted from the original 3D-GS [14] using the loss function,

$$\mathcal{L}_G = (1 - \lambda)\mathcal{L}_1 + \lambda\mathcal{L}_{SSIM} \tag{5}$$

In the second stage, we initialize the audio-guidance parameters of $G_a$ using physical parameters of the warmed-up 3D Gaussians. Subsequently, we learn the implicit material properties for the Gaussian points using gradients obtained in the binauralization task for every training auditory perspective. Optimization of $G_a$ is guided by a combination of the binaural audio reconstruction loss, $\mathcal{L}_m$ and a volume regularization loss, $\mathcal{L}_v$.

$$\mathcal{L}_{G_a} = (1 - \lambda_a)\mathcal{L}_m + \lambda_a\mathcal{L}_v \tag{6}$$

$$\mathcal{L}_m = \mathcal{L}_2(s_m) + \mathcal{L}_2(s_l) + \mathcal{L}_2(s_r) \tag{7}$$

where, $\mathcal{L}_m$ is the summation of the $\mathcal{L}_2$ loss calculated individually between the predicted magnitudes for $s_m, s_l, s_r$ (i.e., mixture, left and right audio channel, respectively) and their respective ground-truth magnitudes. $\mathcal{L}_v = \sum_{i=1}^{N_a} Prod(\alpha_i)$, where $Prod(.)$ is the product of the values of the audio-guidance parameter $\alpha$ of each Gaussian point in the auditory perspective, encouraging the audio-guidance parameters to be small, and non-overlapping.

The physical properties and material-related properties of the new Gaussians are decoded from the learned physical parameters and audio-guidance parameters respectively, in a view-dependent manner on-the-fly.

**Audio-aware point management** The location of points in $G_a$ are initialized from $G$, however since $G$ is optimized using an image reconstruction loss, the initial placement of the points in $G_a$ is not necessarily contributive to an optimal audio guidance. Especially, it has been observed in recent works that 3D-GS is inadequate in densifying texture-less and less observed areas [20, 6, 2].

To address this problem, we propose an error-based point growing policy that populates new points in $G_a$, wherever deemed significant. Specifically, the densification step is interleaved across multiple binauralization forward passes. During each densification step, we compute averaged gradients (across all steps up to the previous consecutive densification step) for each point in $G_a$ that lies in the vicinity, denoted as $\Theta_g$. Points with $\Theta_g > \tau_g$ are deemed as significant, where $\tau_g$ is a pre-defined threshold. Using the original 3D position as a probability distribution function (PDF), we sample new points. The audio-guidance parameters for these new points are obtained by random initialization. Additionally, we regularly employ a random elimination of points to avoid rapid expansion of new points (e.g., every 3000 iterations).

## 4 Experiments

### 4.1 Datasets

We evaluate both a real-world dataset-RWAVS and a synthetic dataset-SoundSpaces.

**Real-World Audio-Visual Scene (RWAVS)** [17] The RWAVS dataset offers realistic multi-modal training samples constituting camera poses, high-quality binaural audios, and images. The dataset consists of 11 indoor and 2 outdoor scenes. Randomly sampling the sound source and listener positions, the authors collected data ranging from 10 to 25 minutes for every scene. For every scene, a data sample includes a set of camera poses (sound source and listener), extracted RGB key frame, followed by one-second binaural audio (as received at the listener position) and one-second mono source audio (as emitted by the sound source). We follow an 80:20 train-validation split for every scene in the dataset.

**SoundSpaces synthetic dataset** [3] The Soundspaces dataset consists of 6 indoor scenes with varying degrees of complexity (2 scenes having a single room with rectangular walls; 2 scenes having a single room with a non-rectangular walls; 2 with multi-room layout). Since the dataset includes room impulse responses (RIR) recorded at receiver/listener positions, we replace the acoustic mask generation block with an RIR prediction block. The listener consists of a stereo listener, with four discrete head orientations among 0, 90, 180, and 270. We follow the same 80:20 train-validation split, as [17] for every scene in the dataset.

### 4.2 Implementation Details

Our complete implementation is based in PyTorch using a single Nvidia A550 GPU. The learning rate for all physical parameters in $G$ is adopted from the original 3D-GS implementation [14]. For the acoustic field network $\mathcal{F}$, we use MLP network with 64 nodes. Hyper-parameters for the binauralizer $\mathcal{B}$, as well as the metrics are adopted from AV-NeRF [17]. For the audio-aware point management, we set $\tau_g$ to 0.0004. Additionally, after every 3k iterations we randomly eliminate points using an outlier removal process (from Open3D [44]) that removes points that have less than 8 neighbors in a sphere less than radius of 0.1.

As a part of the proposed decoupling, we train the 3D-GS model $G$ for 3k iterations, wherein each iteration involves randomly sampling a camera view $V_C$ and optimizing the parameters for $G$. Post this we initialize $\alpha$ with the physical parameters from $G$, and train for 40k iterations, wherein now each iteration involves randomly sampling an auditory perspective $V_A$ and optimizing the parameters for $G_a$ and $\mathcal{B}$.

### 4.3 Results

**Binaural audio synthesis - RWAVS dataset - Table 1** In order to have a fair comparison, we adopt the same metrics - magnitude distance [MAG] [41] (computed in time-frequency domain) and envelope distance [ENV] [26] (computed in time domain), as [17]. Particularly, (1) Mono-Mono duplicates the source audio to create fake binaural audio without modifications; (2) Mono-Energy scales the input audio's energy to match the target, generating stereo audio by duplicating the scaled audio; (3) Stereo-Energy separately scales the input audio's energy for each channel to match the target, then combines them for stereo audio; In addition to the codec baselines above, we also compare with NAF [21], INRAS [34], and AV-NeRF [17]. AV-GS significantly surpasses all prior arts across all scenes, by a significant margin. Particularly compared against AV-NeRF, which utilizes

Table 1: Comparison with state-of-the-art methods on RWAVS dataset.

| Methods | Modality | | Office ↓ | | House ↓ | | Apartment ↓ | | Outdoors ↓ | | Overall ↓ | |
| | Audio | Visual | MAG | ENV | MAG | ENV | MAG | ENV | MAG | ENV | MAG | ENV |
|---|---|---|---|---|---|---|---|---|---|---|---|---|
| Mono-Mono | ✓ | ✗ | 9.269 | 0.411 | 11.889 | 0.424 | 15.120 | 0.474 | 13.957 | 0.470 | 12.559 | 0.445 |
| Mono-Energy | ✓ | ✗ | 1.536 | 0.142 | 4.307 | 0.180 | 3.911 | 0.192 | 1.634 | 0.127 | 2.847 | 0.160 |
| Stereo-Energy | ✓ | ✗ | 1.511 | 0.139 | 4.301 | 0.180 | 3.895 | 0.191 | 1.612 | 0.124 | 2.830 | 0.159 |
| INRAS [35] | ✓ | ✗ | 1.405 | 0.141 | 3.511 | 0.182 | 3.421 | 0.201 | 1.502 | 0.130 | 2.460 | 0.164 |
| NAF [22] | ✓ | ✗ | 1.244 | 0.137 | 3.259 | 0.178 | 3.345 | 0.193 | 1.284 | 0.121 | 2.283 | 0.157 |
| AV-NeRF [17] | ✓ | ✓ | 0.930 | 0.129 | 2.009 | 0.155 | 2.230 | 0.184 | 0.845 | 0.111 | 1.504 | 0.145 |
| **AV-GS (ours)** | ✓ | ✓ | **0.861** | **0.124** | **1.970** | **0.152** | **2.031** | **0.177** | **0.791** | **0.107** | **1.417** | **0.140** |

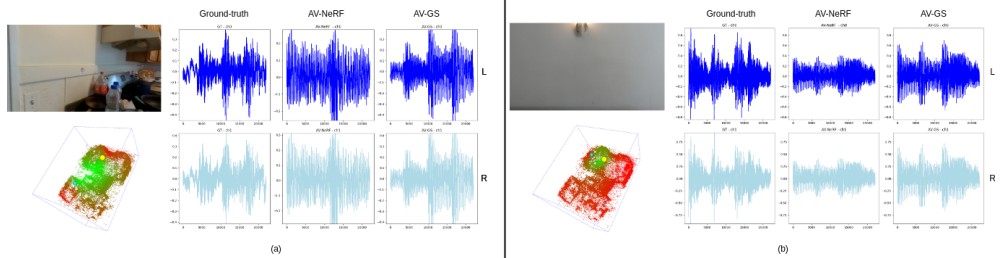

Figure 3: In the presence of (a) complex geometry, and (b) meaningless views, AV-NeRF, when compared to our AV-GS makes errors in binaural synthesis. For both scenarios we showcase the corresponding listener view, used by AV-NeRF, as well as the learned holistic scene representation that is used by AV-GS, and hence unaffected by both scenarios.

audio-visual cues, AV-GS achieves a 5.7% and 3.4% relative improvement in terms of MAG and ENV respectively.

We highlight the qualitative improvement of AV-GS over AV-NeRF in Fig. 3 (and in Section A.1). AV-NeRF extracts visual cues from the listener's view using a 256x256 RGB and depth image (rendered by a pre-trained NeRF) and processed by a frozen ResNet18 [11] model that is trained on ImageNet [7]. When the listener's view includes multiple objects or complex geometries (Fig. 3(a)), or meaningless information (Fig. 3(b)), it leads to sub-optimal visual cue extraction and poor binauralization. In contrast, our AV-GS uses learned representations (combining $\alpha$ of points in $G_a$ w.r.t to the listener and speaker) to extract context at a more holistic level, remaining unaffected by these scenarios.

**RIR generation - Soundspaces dataset - Table 2(a)** Following [17] and [21], we compare AV-GS against classical high-performance audio coding methods: Advanced Audio Coding (AAC) [12] and Xiph Opus [40], applying both linear and nearest neighbor interpolation techniques to the coded acoustic fields. Additionally, we also compare with existing neural methods: NAF [21], INRAS [34], and AV-NeRF [17]. It can be observed that AV-GS outperforms all the prior arts by a significant margin.

Table 2: Comparison with the state-of-the-art and Ablation study on physical parameters.

(a) Comparison with state-of-the-art: Performance on SoundSpaces dataset using T60, C50, and EDT metrics. Lower score indicates a higher RIR generation quality. Opus is an open audio codec [40], and AAC is a multi-channel audio coding standard [12].

| Methods | Audio | Visual | T60 (%) ↓ | C50 (dB) ↓ | EDT (sec) ↓ |
|---|---|---|---|---|---|
| Opus-nearest | ✓ | ✗ | 10.10 | 3.58 | 0.115 |
| Opus-linear | ✓ | ✗ | 8.64 | 3.13 | 0.097 |
| AAC-nearest | ✓ | ✗ | 9.35 | 1.67 | 0.059 |
| AAC-linear | ✓ | ✗ | 7.88 | 1.68 | 0.057 |
| INRAS [35] | ✓ | ✗ | 3.14 | 0.60 | 0.019 |
| NAF [22] | ✓ | ✗ | 3.18 | 1.06 | 0.031 |
| AV-NeRF [17] | ✓ | ✓ | 2.47 | 0.57 | 0.016 |
| AV-GS (ours) | ✓ | ✓ | **2.23** | **0.53** | **0.014** |

(b) Ablation on the choice of physical parameters from $G$, for initializing $\alpha$.

| Parameters | Dimension | Overall ↓ | |
| | | MAG | ENV |
|---|---|---|---|
| $\mathcal{O}$ | 1 | 1.477 | 0.140 |
| $\mathcal{S}$ | 3 | 1.451 | 0.141 |
| $\mathcal{R}$ | 4 | 1.439 | 0.140 |
| $\mathcal{SH}$ | 48 | 1.435 | 0.140 |
| $\mathcal{S}, \mathcal{O}$ | 4 | 1.481 | 0.141 |
| $\mathcal{SH}, \mathcal{O}$ | 49 | 1.466 | 0.141 |
| $\mathcal{S}, \mathcal{SH}$ | 51 | 1.549 | 0.143 |
| $\mathcal{SH}, \mathcal{R}$ | 52 | **1.417** | **0.140** |
| $\mathcal{S}, \mathcal{SH}, \mathcal{O}$ | 52 | 1.500 | 0.141 |
| $\mathcal{SH}, \mathcal{R}, \mathcal{O}$ | 53 | 1.454 | 0.141 |
| $\mathcal{S}, \mathcal{SH}, \mathcal{R}, \mathcal{O}$ | 56 | 1.496 | 0.141 |

Table 3: Comparing for efficiency in terms of the inference time with the existing AV-NeRF using multiple perspective views. Please note the inference time includes the time for rendering the view from V-NeRF, extracting features through AV-Mapper ([17]) and then rendering the binaural audio from A-NeRF.

| Methods | Time ($\downarrow$) Inference | Overall ($\downarrow$) MAG | ENV |
|---|---|---|---|
| AV-NeRF - 1 view | 1.4s | 1.504 | 0.145 |
| AV-NeRF - 2 views | 2.9s | 1.495 | 0.145 |
| AV-NeRF - 4 views | 7.6s | 1.482 | 0.144 |
| AV-GS (ours) | **0.08s** | **1.417** | **0.140** |

**Improved efficiency** Decoding $\alpha$ on-the-fly helps AV-GS retain the efficiency advantage of having an explicit point based representation over learning an implicit representation of the visual 3D scene. In contrast, AV-NeRF trains an additional NeRF model, whose outputs are used to render the listener's view which is the condition for their binauralizer module. In Table 3, we compare the inference time for a single scene (House scene 1) from the RWAVS dataset. Although the original implementation of AV-NeRF proposed a single view in the direction of listener's viewing direction (used as the listener's context), we modify their implementation in which the V-NeRF renders 2 and 4 perspective views with a field of view of 90° for each receiver's position. It can be observed, although using the additional views, helps provide additional local context around the listener (hence improving the binauralization performance), but only at the cost of increased inference time. AV-GS on the other hand requires least rendering time, while providing the best binauralization performance. It is important to note that, although we train the 3D-GS for only an initial warm-up stage of 3k iterations, we can still jointly train both $G$ and $G_a$ on the corresponding camera view $V_C$ and the auditory perspective $V_A$ respectively. By doing so, AV-GS would be able to synthesize both binaural audio as well the visual view at the unseen target (novel) viewpoint, without any increase in the training time than that of as shown in Table 3 (last row). In this work, we instead focus on only the binaural audio synthesis and hence train only $G_a$ in the second stage.

Although we train the 3D-GS for just an initial 3k iterations, we can still train both $G$ and $G_a$ together for the corresponding camera view $V_C$ and auditory perspective $V_A$, such that the trained AV-GS is able to synthesize both the binaural audio and the visual view from a novel viewpoint , thereby making it a fair comparison with AV-NeRF [17]. However, in this work, we focus solely on binaural audio synthesis and only train $G_a$ in the second stage.

## 4.4 Ablation study

We use the binaural audio synthesis task for performing our ablation study and report average performance across all scenes in the RWAVS dataset.

**Initialization of $\alpha$.** 3D-GS learns a point-based representation ($G$) of the 3D scene such that each point in the explicit representation is represented as a 3D Gaussian ellipsoid to which physical attributes like position $\mathcal{X}$, quaternion $\mathcal{R}$, scale $\mathcal{S}$, opacity $\mathcal{O}$ and Spherical Harmonic coefficients ($\mathcal{SH}$) representing view-dependent color are attached. As discussed above, we initialize the audio-guidance parameter $\alpha$ using the learned physical parameters from $G$. Table 2(b) shows an in-depth study of the effect on the binauralization performance when choosing different parameters from $G$ for the initialization. It is evident that $\mathcal{SH}$ and $\mathcal{R}$, when considered individually as well as when combined, are crucial parameters that help provide an overall better binauralization performance.

**Size of the vicinity** The learned Gaussian points $G$ representation normally consists of millions of points, and using all the points for computing the condition for the binauralization task will result in a huge computational overload. Moreover, intuitively, points near the listener and speaker location in the 3D space (*i.e.* vicinity) are more contributive to determining the local geometry and material-related characteristics, and greatly influence the sound field. Empirically in Table 4a, we find that using 15 percentile of the points in the vicinity (listener and speaker considered separately) yields the best performance. Using an extremely high percentile for capturing the vicinity generally helps, but only to a certain extent, since it increases the risk of adding unnecessary information. Moreover, it is important to note that in scenes with smaller room sizes or when the listener and speaker are

Table 4: (a)Ablation on the size of vicinity w.r.t the listener and sound source position. (b) Effect of audio-aware point management.

(a) Percentile-k denotes the top $k$ % points nearest to the listener and sound source.

| Percentile | Overall ↓ | |
|---|---|---|
| | MAG | ENV |
| 5 | 1.482 | 0.143 |
| 10 | 1.454 | 0.141 |
| 15 | **1.417** | **0.140** |
| 20 | 1.424 | 0.141 |
| 25 | 1.428 | 0.141 |

(b) SfM: Structure-from-motion [33] (*i.e.*camera calibration), $G$: warmup using 3D-GS [14]

| Initialization | Point management | Overall ↓ | |
|---|---|---|---|
| | | MAG | ENV |
| $SfM$ | ✗ | 1.668 | 0.147 |
| | ✓ | 1.552 | 0.143 |
| $G$ (3D-GS) | ✗ | 1.481 | 0.141 |
| | ✓ | **1.417** | **0.140** |

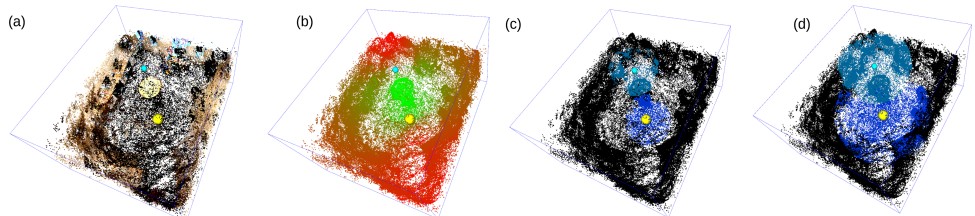

Figure 4: Ablation on the size of vicinity w.r.t the listener and sound source position. Percentile-k denotes the top $k$ % points nearest to the listener and sound source are considered in computing the scene context. (a) RGB color (from $G$), (b) learned $\alpha$ (from $G_a$), (c) 5% percentile vicinity, (d) 25% percentile vicinity.

placed nearby, a larger vicinity capture might lead to overlapping points thus adding to redundant information (see Fig. 4 (d)).

**Effect of audio-aware point management.** AV-GS infers local geometry from an explicit point-based prior and learns additional audio-specific parameters for every point. The local geometry can be inferred either directly from the sparse points produced during camera calibration using Structure-from-motion (SfM) [33] or warming up a 3D-GS model on the SfM points. From Table 4b it is evident that audio-aware point management helps improve the binauralization performance irrespective of the adopted point initialization approach. Texture-less regions such as walls, doors, etc are the regions with maximum sound absorption and diversion and hence having optimal point density promotes better modeling of sound propagation.

## 5 Conclusion

In this work, we present AV-GS, a holistic scene representation learning approach for conditioning novel view acoustic synthesis. The core of AV-GS lies in the proposed decoupled modeling of 3D scene geometry and material-related characteristics of scene objects for sound propagation, enabled by the introduction of additional audio-guidance parameters per point within an acoustic field network. We show that our approach leverages audio-aware Gaussian point positioning to improve the binauralization performance, on real world as well as synthetic 3D scenes, significantly in comparison to prior art alternatives.

**Limitations** Although AV-GS achieves a new state-of-the-art on the NVAS task by learning a holistic scene representation with geometry-aware and implicit material-aware characteristics, it faces some challenges. Like AV-Nerf, AV-GS currently learns a separate representation for each scene, posing a significant challenge for generalizability and transferability across multiple scenes, which is crucial for improving efficiency and reducing computational demands. Also, as observed in Table 1, the performance drops when the scene size increases (e.g., from Office to House), due to the difficulty in learning representations for larger, complex geometries with multiple rooms. An interesting research direction would be to model individual rooms using separate 3D-GS representations and then learn a transfer function across rooms.

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

# A Appendix / supplemental material

**Broader impact** Potential applications of AV-GS lie in virtual reality, augmented reality, and immersive audio experiences, where accurate and efficient binaural audio rendering can significantly enhance user experience. However, enhanced acoustic synthesis technology could be used unethically in surveillance or privacy-invasive applications, where binaural audio could be synthesized to eavesdrop on conversations or create misleading audio experiences.

## A.1 Direction and distance awareness showcased by AV-GS

Fig. 5 highlights the distance-awareness exhibited in the binaural audios synthesized by AV-GS. The top row shows the corresponding RGB frame or the listener's view, followed by AV-GS's learned holistic scene representation in the next row. The two rows show a plot of the rendered binaural audios (left and right respectively) with time along the x-axis and amplitude of the audio along the y-axis. (Please note that in the second row we slice the points in $G_a$ for along the vertical axis, essentially removing the points from the ceiling, in order to provide better visibility.) With the increase in the distance of the listener (blue sphere) from the sound source (yellow sphere), AV-GS is able to synthesize binaural audios with decreasing amplitude across both the left and right audio channels.

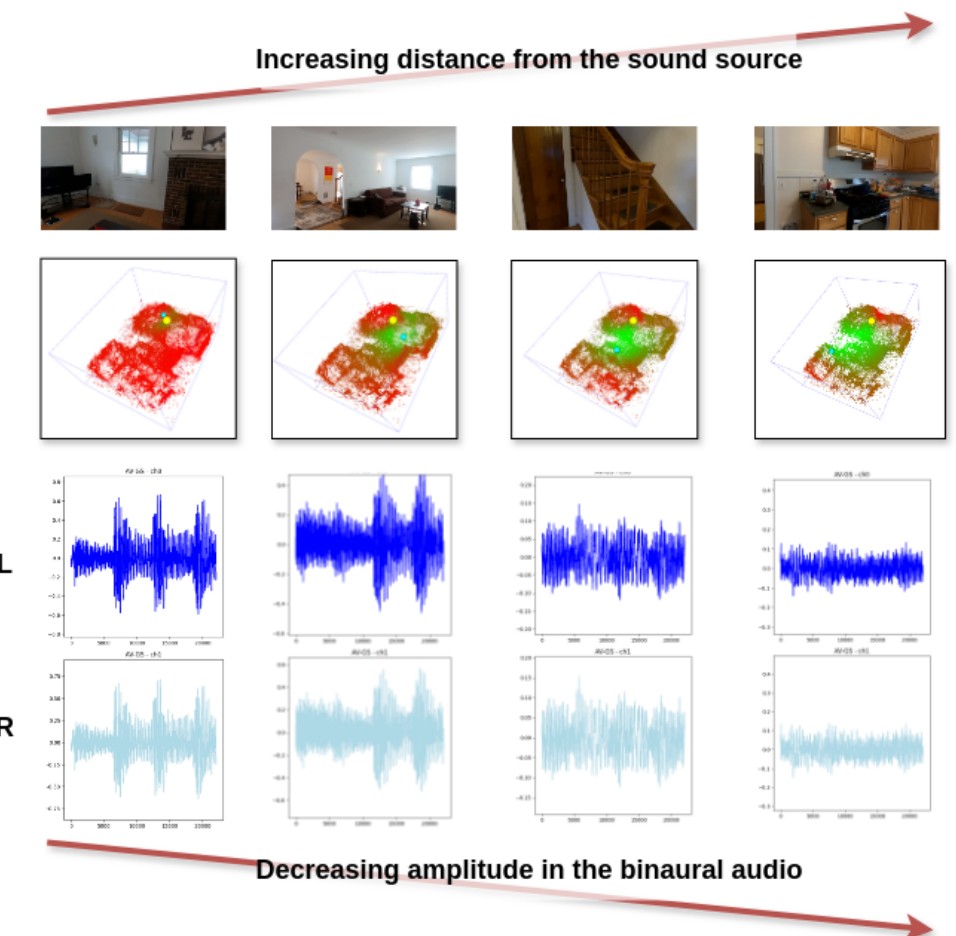

Figure 5: Distance aware audio rendering. As the distance from the sound source increases the amplitude of the synthesized binaural audio decreases. Yellow sphere - location of the sound source, blue sphere - location of the listener.

Fig. 6 shows the direction-awareness showcased by AV-GS. In (a) the synthesized left channel has a higher magnitude compared to the right channel, owing to the proximity of the listener's left ear to

the sound source. Vice-versa, in (b) the right channel has a higher amplitude compared to the right channel owing to the proximity of the listener's right ear to the sound source.

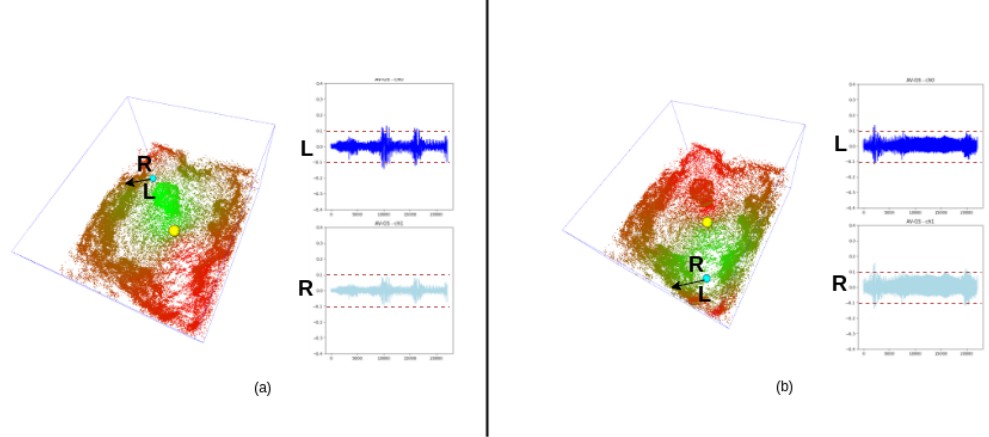

Figure 6: Direction aware audio rendering. Relative to the viewing direction of the listener w.r.t to the sound source AV-GS synthesizes binaural audios with varying amplitude levels in the left and right audio channel. The arrow on the listener (blue sphere) represents the viewing direction, and the yellow sphere depicts the sound source.

## A.2   Binauralizer

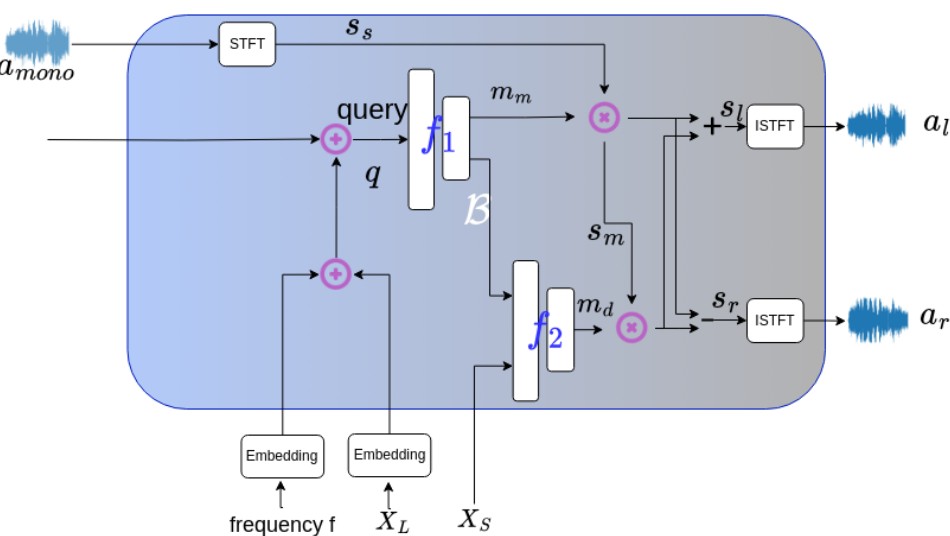

Figure 7: Overview of the binauralizer module, $\mathcal{B}$ (with modifications from [17]).

We adopt the architecture of binauralizer from [17] with some modifications to input the learned scene context from our acoustic field network $\mathcal{F}$ (see Fig. 7).

Embedding blocks take the normalized location position $X_L$ (only 2D x and y coordinates).

The main components include two Multilayer Perceptrons (MLPs), each comprising four linear layers with an additional residual connection. The width of each linear layer, is set to 128 for the RWAVS dataset and 256 for the SoundSpaces dataset. All linear layers are followed by ReLU activation layers, except for the last layer, where the ReLU activation is replaced with the Sigmoid function. The first MLP takes the listener's position (x, y) and the frequency $f \in [0, F]$ as input, where $F$ represents the number of frequency bins. It predicts a mixture mask $m_m$ for the given frequency $f$ and generates a

feature vector with $c$ channels. Prior to feeding them into the MLP, we apply positional encoding to the listener's position (x, y) and the frequency $f$. We set the maximum frequency used for positional encoding as 10. A coordinate transformation [17] to project the listener's direction $\theta$ into a high frequency space. We concatenate the transformed listener's direction and the feature vector, and feed it into the second MLP. The second MLP is appended with a Sigmoid layer and a scaling layer, ensuring that the difference mask $m_d$ estimated by the second MLP falls within the range of $-1, 1$. For each frequency query $f$, estimates two masks: $m_m$ and $m_d$, both of which are scalars. We iterate over all frequencies $f \in [0, F]$ to obtain the complete masks $m_m$ and $m_d$.

For validating AV-GS on the room impulse response generation task using the Soundspaces dataset, following [17], we drop the mixture mask, and predict the impulse response directly for a corresponding time input, using the second MLP. We show these modifications in Fig. 8.

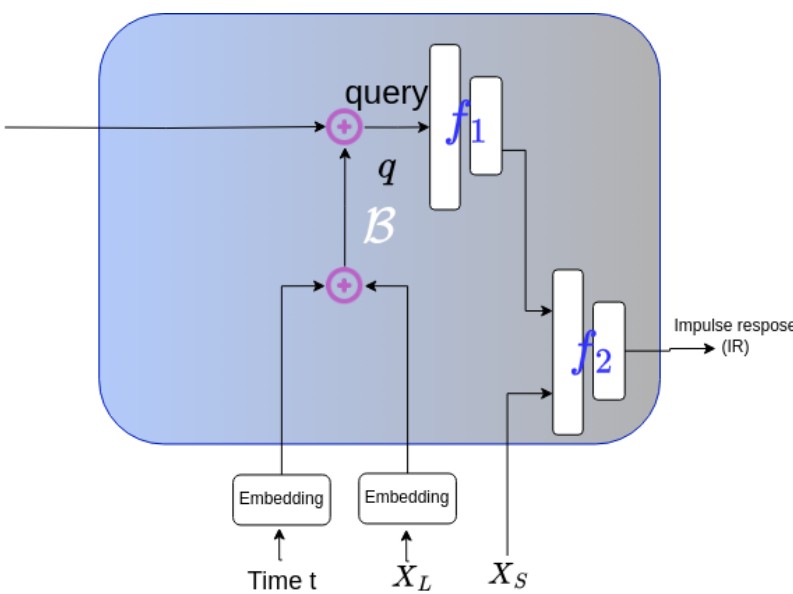

Figure 8: Changes to $\mathcal{B}$ for the RIR generation - Soundspaces dataset (scores reported in Table 2(a)).

