# OpenReview forum: "AV-GS: Learning Material and Geometry Aware Priors for Novel View Acoustic Synthesis"
_NeurIPS.cc/2024/Conference — NeurIPS 2024 poster_

### Official Review · Reviewer_DaxK · 2024-07-10

**Soundness:** 3
**Presentation:** 2
**Contribution:** 3
**Rating:** 6
**Confidence:** 5

**Summary:**

This paper proposes a novel view acoustic synthesis approach based on 3D Gaussian Splatting scene representation. The framework consists of a 3D GS model, acoustics field network, and audio binauralizer. It first trains a 3D GS model to capture scene geometry. Then they use attributes from each Gaussian to initialize a learnable acoustic point representation, and then average the feature obtained from nearby Gaussian points for acoustic modeling.
Through the experiments on the RWAVS and Soundspaces dataset, the authors demonstrate that the proposed method achieves state-of-the-art performance. They also perform the ablation study to ablate some components to demonstrate their technical advance.

**Strengths:**

- The paper brings up a clever way to leverage point-based scene representation from 3DGS for acoustic modeling and the results are promising.
- The authors perform thorough experiments and ablations to demonstrate its technical advance. The proposed method outperforms baselines on the quantitative evaluation.

**Weaknesses:**

- The reviewer thinks there are several issues with the writing:
  - First, **the method part isn't clearly stated**. It takes a long time for the reviewer to fully understand the method. The method is simple, using pretrained 3DGS to initialize a learnable acoustic point representation. While the method figure doesn't quite help with understanding.
  - The reviewer strongly feels the paper is **overclaimed**. It claims it learns "holistic geometry-aware material-aware scene representation" while the experiments are only for acoustic modeling results. The author needs to use experiments to support this claim. One thought the reviewer has to visualize the PCA feature of learnable acoustic points to see if it has some cues. Otherwise, the reviewer is only convinced that this paper proposed a better acoustic modeling approach that leverages the GS features.
  - The major experimental **results for the baselines are directly brought from AV-NeRF and INRAS papers** while the authors don't mention this in the paper. Also, the authors need to confirm if they have guaranteed the same training and test split and evaluation code.

- The reviewer has a question on the Position-guidance G condition. The authors use unit vectors which get rid of distance information. The reviewer is confused as to why remove the distance. Would it be an important cue?
- In the paper [5] Real acoustic field, the authors found out that energy decay loss shows a powerful improvement on those energy-based metrics. Did authors consider using this loss to improve?

**Questions:**

I have mentioned my concerns and questions in the weaknesses part and I hope the authors could address them during the rebuttal.

A suggestion to the authors for paper writing: save the figures as PDF instead of screenshots or png.

**Limitations:**

Yes, the authors adequately addressed the limitations

---

> ### Author Rebuttal · Authors · 2024-08-06
>
> ## Weakness
>
> **W1: Method part isn't clearly stated**
> We will re-draw Figure 2 (Overview of our proposed AV-GS) to make it more clear and straightforward. And update all figures to PDF.
>
> **W2: PCA feature of learnable acoustic points**
> (The PCA plot and the correlation matrix discussed below have been provided in the PDF attached above, in the common "Author Rebuttal")
> For RWAVS scene 6, we analyze alpha by plotting its first two principal components and applying k-means (k=3) clustering, to project colors onto G_a. It can be observed that most of the objects are grouped by the same color. Next, we manually match scene objects (e.g., fireplace, walls) with material labels (e.g., 'brickwork', 'solid wall' respectively) from an absorption database [6]. Using these absorption coefficients, we assign absorption vectors (Dim: [1x6]; coefficients for 6 frequencies from [6]) to every point that corresponds to the object (matched with the label). A correlation matrix of these absorption coefficients, across the points (grouped by clusters), shows high self-correlation within clusters and lower cross-correlation across clusters, i.e., the points within the same clusters have similar absorption values, and less correlated absorption across the clusters. While these clusters do not explicitly highlight fine-grained material properties such as Young's modulus, they provide implicit material cues for binauralization.
>
> Additionally, we demonstrate the effect if alpha is frozen during our training (initialized using SH from vanilla 3D-GS). As shown in the table below, constraining alpha to SH results in a significant drop in binarualization.
>
> | Alpha- trainable | MAG (↓) | ENV (↓) |
> |------------------|---------|---------|
> | No               | 1.584   | 0.147   |
> | Yes              | 1.417   | 0.140   |
>
> Combined with the PCA/correlation matrix above, and this experiment, we hypothesize that the learned alpha is able to pick up implicit material properties on top of SH to aid binauralization. We agree that alpha captures an implicit, rather than explicit properties like Young's modulus, material representation derived from raw RGB images and will clarify this in the revised version.
>
> **W3: confirm if same train-test split and evaluation code as baseline**
> Thank you for pointing this out. We adopt the same training and test split and the evaluation code provided by AV-NeRF (available on Github) and hence making the comparison fair.
> Please note AV-NeRF provides train and test json files, and we adopt the same dataloader from their code, which makes the comparison fair.
>
> **W4: Position-guidance. Isn't distance an important cue?**
> Great question! We found using only the view information for position guidance suffices for the model performance. We found that this normalization in Eq. (3) provides numerical stability during training.
> With regards to the distance being an important que, please note that the position embedding of the listener location is already a part of the binauralizer, please check the Fig. 7 in A.2
> We skip the arrow (showing X_L/X_S into the binauralizer) in the Fig 2, to maintain brevity and not deviate focus of the novel contributions. Instead we tried to highlight this in our Binauralizer Fig. 7 in A.2.  We will update this in the revised version.
>
> **W5: Does energy decay loss help?**
> Thank you for pointing this out. We adopted the decay loss in addition to our proposed losses, however we did not find any improvement. (please note, for the RWAVS dataset, we compute the l1 loss between the decay curves for the GT and the predicted spectrograms of the binaural audio).
>
> Following RAF[7] we also ablated the weight of the decay loss in the overall loss computation. However, in contrast to RAF's findings where decay loss improves temporal domain metrics, but worsens temporal-frequency domain metrics (STFT error), we find that it worsens both temporal (Envelope distance (ENV)) and temporal-frequency domain metrics (spectrogram magnitude (MAG).
>
> | Lambda (decay weight) | MAG (↓) | ENV (↓) |
> |-----------------------|---------|---------|
> | 0.0                   | 1.417   | 0.140   |
> | 1.0                   | 1.481   | 0.142   |
> | 2.0                   | 1.510   | 0.142   |
> | 3.0                   | 1.486   | 0.141   |
>
>
> References:
> [1] Gao, Ruohan, and Kristen Grauman. "2.5 d visual sound." CVPR'19
>
> [2] Luo, Andrew, et al. "Learning neural acoustic fields" NeurIPS'22
>
> [3] Su, Kun, Mingfei Chen, and Eli Shlizerman. "INRAS" NeurIPS'22
>
> [4] Liang, Susan, et al. "AV-NeRF." NeurIPS'23
>
> [6] C. Kling. Absorption coefficient database, Jul 2018.
>
> [7] Chen, Ziyang, et al. "Real acoustic fields" CVPR'24
>
> [8] Tang, Zhenyu, et al. "GWA" SIGGRAPH'22

---

> > ### Comment · Reviewer_DaxK · 2024-08-10
> >
> > Thank you for the response. The PCA results show that the model learns material-related properties, and the explanation of distance information clarifies my concerns. I’ll raise my score to ‘weak accept’ and hope these changes are reflected in the revised version.

---

> > > ### Author Response · Authors · 2024-08-10
> > >
> > > We thank the reviewer for recommending acceptance. We appreciate this discussion and will incorporate it in our final version.

---

### Official Review · Reviewer_RHwj · 2024-07-10

**Soundness:** 3
**Presentation:** 2
**Contribution:** 2
**Rating:** 5
**Confidence:** 4

**Summary:**

This paper proposes a new audio-visual Gaussian Splatting model for novel view acoustic synthesis. The proposed method explicitly models scene geometry and learns a point-based scene representation with an audio guidance parameter on locally initialized Gaussian points that takes the space relation from the listener and sound source into consideration. This work is the first that applies gaussian splatting to the novel view acoustic synthesis problem, and experiments on two datasets demonstrate the effectiness of the proposed method compared to prior work.

**Strengths:**

- The idea makes sense, and I was expecting gaussian splatting to be applied to this problem by somebody sooner or later, and this paper is indeed doing that.

- Generally, the paper makes good attempts to apply Gaussian Splatting to this recently proposed task, a direct extension on how it is used for visual rendering. The method is clearly formulated and defined with informative notations.

- Experiments are conducted on two datasets following prior work AV-NeRF [17], and it shows noticeable gains compared to prior baselines.

- Many alation studies are performed including ablation on the size of vicinity w.r.t the listener and sound source position, ablation on the choice of physical parameters, which is extensive.

**Weaknesses:**

- There is a strong claim that the proposed model is learning material-aware priors, jointly modeling 3D geometry and material characteristics, holistic scene geometry and material information, etc. Then, I would actually expect materials are explicitly modeled, such as acoustic properties of surfaces in the form of some material parameters (e.g., absorption coefficients, Young’s modulus, etc.). However, all that is modeled is just a parameter alpha that claims to encapsulate material-specific characteristics of the scene. This is somewhat misleading as there is nothing constrains alpha to be physics-based or to be correlated with some material parameters. It's likely that it’s just learning some scene representation that may be more helpful for acoustic rendering.

- Also the so-called acoustic field network is also nothing about audio by itself. By acoustic field network, I was thinking the method is explicitly modeling sound propagation by taking into account the material properties of surfaces and the geometry of the space. The audio binauralizer is basically following prior work AV-NeRF [17], and what changes is what the audio binauralizer conditions on. In AV-NeRF, it’s scene features from NeRF and in this work it’s the context scene features from gaussian splatting.

- There were no qualitative examples given in supplementary materials, so it's really hard to appreciate the improvement that this method leads to.

- There are also many typos or grammar mistakes here and there. For example:
1. L45, 3D Gaussian splatting-based *methods*?
2. Related work, mixed use of past tense and present tense.
3. L125, a brief ??

**Questions:**

- See questions in the weakness section above.

- Also, for the SoundSpaces synthetic dataset, why it only contains 6 indoor scenes. There are more scenes in the dataset, and it would be good to explain the reason why only these six are not used.

- Is there any way to interpret the learned audio-guidance parameter to show that it’s material related?

**Limitations:**

Yes some limitations are discussed in the end, though some might be missing as discussed above.

---

> ### Author Rebuttal · Authors · 2024-08-06
>
> ## Weakness
>
> **W1: Explicit modeling of alpha**
> We appreciate the reviewer's insight regarding the explicit modeling of alpha.
>
> *"there is nothing constraining alpha to be physics-based or correlated with material parameters"*
>
> While we acknowledge the importance of deriving alpha from material properties input, we would like to clarify that AV-GS is motivated by the latest trend towards scalability and adaptability for real-world scenes like RWAVS[4] and Real acoustic field[7], where only RGB frames are available – a more practical and least constrained setting. In contrast, other works [5] [8] using absorption coefficients rely on synthetic scenes and pre-existing absorption databases, matching object labels with material descriptions which are available for synthetic scenes but typically unavailable in real world scenarios as considered in our work.
>
> Please consider that currently, anyone with a phone and a binaural microphone can capture all input data that is required to train an AV-GS model (~4 min of recording for a single room scene) - something not possible if explicit material properties are required for model training.
>
> We agree that alpha captures an implicit, rather than explicit, material representation derived from raw RGB images and will clarify this in the revised version.
> (Also, please see Q2 below, where we demonstrate that alpha, although implicitly learned, is related to material properties.)
>
> **W2: Explicit modeling of acoustic field network**
> Explicit sound propagation modeling is indeed valuable, but our acoustic field network design is intentionally non-trivial. We learn acoustic properties on 3D-GS points, accumulating points within an empirically validated vicinity of the listener and source. We fuse implicit material (alpha) with learned position guidance, using point view directions relative to listener/source. This is further facilitated by audio-aware adaptive density control to priortise point density across texture-less regions -something vanilla 3D-GS lacks, but is important for NVAS. We agree explicit beam-tracing within audio-guided 3D-GS is an interesting future research direction.
>
> Regarding Binauralizer, as mentioned in L166, it is not a part of our contribution. It was motivated in [1] and adopted by AV-NeRF[4]. We further improve it by deriving holistic conditions from 3D scenes, specifically- learning implicit material properties, fusing with position guidance, and adaptively densifying the point-based representation w.r.t audio reconstruction loss.
>
> **W3: Qualitative audio samples**
> Because of rebuttal instructions regarding not sharing links, we have provided the AC with an anonymized link to the audio samples. Apologies for the inconvenience. We will also make our code repo and all rendered samples publicly available post the review period.
>
> **W4: Typos**
> Apologies! we will address all the typos in our revised version.
>
> ## Question
>
> **Q1: only 6 scenes from Soundspaces?**
> We chose six indoor scenes from the SoundSpaces dataset to ensure consistency and valid fair comparisons with previous works, specifically NAF (NeurIPS'22), INRAS(NeurIPS'22), and AV-NeRF(NeurIPS'23), which established benchmarks using these six scenes. They represent a diverse range of environments: (1) Two single rooms with rectangular walls, (2) Two with non-rectangular walls, and (3) Two multi-room layouts.
> Currently both RWAVS and soundspaces-synthetic are publicly available, and hence used for our fair and consistent validation. We can incorporate this into our revised version with similar data (binaural audio and RGB pairs) for other scenes.
>
> **Q2: Interpretability of audio-guidance parameter**
> (The PCA plot and the correlation matrix discussed below have been provided in the PDF attached above, in the common "Author Rebuttal")
> For RWAVS scene 6, we analyze alpha by plotting its first two principal components and applying k-means (k=3) clustering, to project colors onto G_a. It can be observed that most of the objects are grouped by the same color (3 clusters - orange, green, blue). Further, we also manually match scene objects (e.g., fireplace, walls) with material labels (e.g., 'brickwork', 'solid wall' respectively) from an absorption database [6]. Using these absorption coefficients, we assign absorption vectors (Dim: [1x6]; coefficients for 6 frequencies from [6]) to every point that corresponds to the scene object matched with the material label from [6]. A correlation matrix of these absorption coefficients, across the points (grouped by clusters), shows high self-correlation within clusters and lower cross-correlation across clusters, i.e., the points within the same clusters have similar absorption values, and less correlated absorption across the clusters. While these clusters do not explicitly highlight fine-grained material properties such as Young's modulus, they provide implicit material cues for binauralization.
>
> Additionally, we demonstrate the effect if alpha is frozen during our training (initialized using SH from vanilla 3D-GS). As shown in the table below, constraining alpha to SH results in a significant drop in binarualization.
>
> | Alpha- trainable | MAG (↓) | ENV (↓) |
> |------------------|---------|---------|
> | No               | 1.584   | 0.147   |
> | Yes              | 1.417   | 0.140   |
>
> Combined with the PCA/correlation matrix above, and this experiment, we hypothesize that the learned alpha is able to pick up implicit material properties on top of SH to aid binauralization.
>
> References:
> [1] Gao, Ruohan. "2.5 d visual sound." CVPR’19
>
> [4] Liang, Susan, et al. "AV-NeRF." NeurIPS’23
>
> [5] Ratnarajah, Anton "Listen2Scene" IEEE VR’24
>
> [6] C. Kling. Absorption coefficient database, Jul 2018.
>
> [7] Chen, Ziyang, et al. "Real acoustic fields" CVPR’24
>
> [8] Tang, Zhenyu, et al. "GWA" SIGGRAPH’22.

---

> > ### Comment · Reviewer_RHwj · 2024-08-11
> >
> > Thanks for the rebuttal and additional results, which helps address many of my concerns. The new PCA plot is especially helpful, and I would encourage the authors to include it in the main paper to demonstrate the correlation of alpha with real material properties. The authors are suggested to revisit this strong claim in the paper "the proposed model is learning material-aware priors", and make appropriate adjustments and provide supporting resutls where needed.
> >
> > Regarding the anonymized link to the audio samples only shared with AC, I kindly request the AC to confirm and double check the audio results are meaningful. If that is confirmed, I am happy to raise my rating and fine with the paper getting accepted considering the interesting idea and task proposed in the paper.  The qualitative audio samples are also important to be included as part of the paper to help readers appreciate the task and the gains of the proposed method.

---

> > > ### Author Response · Authors · 2024-08-11
> > >
> > > We thank the reviewer for the discussion and for increasing the score towards acceptance. We will update the final version with these details and mention the learned implicit material properties.

---

### Official Review · Reviewer_hPZr · 2024-07-10

**Soundness:** 3
**Presentation:** 2
**Contribution:** 3
**Rating:** 7
**Confidence:** 4

**Summary:**

This work proposes an efficient and performant method for novel view acoustic synthesis using Gaussian Splatting representations of 3D scenes. The proposed model integrates geometry and material conditioning, and offers a way to bridge the gap between visually-useful 3DGS representations and acoustically useful ones, using “audio-guidance” which considers the local environment in Experiments show this outperforms existing NeRF-based approaches while being more efficient.

**Strengths:**

Overall, I think this work makes a very useful contribution. Bridging Gaussian Splatting representations with audio rendering is likely to be of significant interest to the community, and the quantitative results and efficiency seem strong. The experiments and comparisons are helpful, and give me confidence that these results are quite robust.

**Weaknesses:**

The value of $\tau_g$ seems somewhat arbitrarily defined, and the range of tested vicinity sizes is somewhat narrow; given the efficiency of the method. It’s hard to extrapolate what the performance would be with more context (i.e. would it keep going down?). Most of my other suggestions are relatively minor, about the writing and presentation:
- Abstract: “at a 3D scene” -> for clarity, should this be “within” a 3D scene?
- Introduction: “visual rendering of 3D scenes without spatial audio (i.e., deaf)” -> this seems like an inaccurate characterization. Deafness is not about rendering, but about sensation. Spatiality is also not clearly related to this.
- Introduction: “realistic binaural audio synthesis is challenging, since the wavelengths of sound waves are much longer” -> this is not the only reason that this is challenging. Even if you use a simplified (e.g. ray-tracing) model, which would be inaccurate at lower frequencies due to wave-like behavior etc., you would still need a good model of the environment for physics-based rendering, or a large number of spatial measurements for a data-driven approach.
- Introduction: “…modeling 3D geometry and material characteristics of the visual scene objects to instigate direction and distance awareness…” -> while geometry has a direct impact on “direction and distance”, material impacts spatial perception indirectly (through absorption, reflection, and diffusion). Could this be rephrased more clearly?
- Related work: “Anton and Dinesh” should maybe be “Ratnarajah and Manocha”, i.e. last names of authors?
- Related work: Under “Geometry and material conditioning”, the final statement differentiates the current work from prior work by pointing out the non-reliance on scene and geometry inputs, but doesn’t clarify why this is useful. E.g. it could be pointed out that, in the real world, such inputs are often not available, so this work allows generalizing to such settings.
- Figure 2: “Differential Rasterizer” -> “Differentiable Rasterizer”?
- Figure 4: Could there be a subfigure showing 15%, since that seems to give the best overall performance? It would be helpful context, since otherwise the reader needs to interpolate between (c) and (d).
- Section 4.4: “intuitionally” -> “intuitively”?

**Questions:**

- Is $S$ modeled as an omni-directional point source? I assume so because point source is the typical assumption, and an emission direction isn’t given (vs. listener direction d), but it’s worth clarifying.
- I think the notation here could be clearer. For example, $V_C$ and $V_A$ don’t seem used, and it’s not clear to me what their role is? E.g. why is the observation $O_p$ = ($V_C$, $V_A$) instead of ($I$, $a_{bi}$)? If the pose $p$ captures the position and orientation, what do $V_C$ and $V_A$ add on top of this representation? Why is $I$ upper-case and $a$ lower-case here, when they both represent observations? Why is $a_{bi} = \{a_l, a_r\}$ notated as a set, but other sets are notated as tuples? Additionally, shouldn’t $k_l$ be $k_L$, similar for $S$?
- Conceptually, there is a tension between considering the global scene context, as the paper argues for, and dropping points outside the local vicinity of the source and listener (since sound reflections can easily include points from other regions, in principle). Could the authors comment on this? It’s interesting that the optimal vicinity size is not the largest one tested. Is there some intuition for why this might be the case, given that it offers more information? Currently, the reasoning seems to be that this contains unnecessary information, but the useful information seems to be a subset of this?
- The audio rendering task is cast as a binauralization task. I understand that this is practically necessary, i.e. the input is mono and output is stereo binaural. However, wouldn’t many of the problems in this paper still exist in the monaural listener context as well? E.g. compared with a binauralization task purely based on HRTFs or simplified room geometries (e.g. Richard et al., ICLR 2021), where the context $C$ is not as rich. Could this be clarified?
- How was the $\tau_g$ value decided?

**Limitations:**

I think the authors have fairly represented the limitations. The broader impact statement is a little bit less clear to me, i.e. I’m not sure I understand the surveillance applications. Could this be clarified? I would be interested for the authors to consider the potential impact of the limitations listed. For example, the limitation of rendering larger scenes; what practical negative impacts could this have in deployment scenarios?

---

> ### Author Rebuttal · Authors · 2024-08-06
>
> ## Weakness
>
> **W1: Value of T_g**
> T_g is inspired by vanilla 3D-GS paper (which empirically found a threshold of 0.0002) and consistently our threshold of 0.0004 was determined through empirical testing.
>
> In 3D-GS optimization (trained for NVS), the point's gradient is compared with the T_g threshold, and is decided whether the point should be cloned/split or not. Particularly in this case the gradients correlate to ability of the points to contribute to the visual geometry of the scene when they are splatted in a particular camera-view direction. In our NVAS case, the gradient of the points correlate to the contribution of the point in providing an acoustic guidance when generating binaural audio for a given {X_L, X_S} pair. Please note NVAS is not limited to only the camera-view direction, but rather vicinity of the listener and sound source.
>
> Our empirical choice is in-line with 3D-GS, and also intuitively higher -because we adopt a dual stage training, wherein the rough geometry is already provided to 2nd stage (Structure-from-Motion points are provided to the first stage), and comparatively considered for a much different sample of points (volumetric camera view Vs listener and source vicinity). A higher value helps to save compute cost.
>
> Given a reasonable memory budget, ideally one would want to empirically decide on the threshold for every different vicinity size.
>
> **Minor comments** We sincerely appreciate the discussion regarding general acoustics, we will incorporate this in our next version with the 15% vicinity plot. We will relate our Intro section to the wave-based (good for lower frequency bands --> resulting in diffusion, scattering .. i.e.  wave properties) and Geometric acoustic-based methods (good for higher frequencies --> behaves like rays --> ideally specular reflections .. i.e. ray properties).
>
> ## Question
>
> **Q1: S omni-directional?**
> Right, the sound source is omni-directional, we adopt the same dataset as AV-NeRF [4]. We will mention this explicitly in our revised version.
> To extend AV-GS for directional sources, directional components can be added (perhaps weighing the vicinity sphere in the direction).
>
> **Q2: Notations**
> Sincere apologies for the confusion.
> V_C and V_A: We tried to use V to represent generic views; V_C - visual and V_A - audio modality.
>
> V_C = \{I | p\}; V_A = \{a_{bi} | p, a_{mono}\}
>
> For the current scope of input used by AV-GS, you are right in saying, V_C = I and V_A = a_bi. However, in a more generalized NVAS setting where one may supplement the RGB with a depth view, in which case V_C = I+D (RGB + Depth).
>
> I and a: As mentioned these are both observations, however, 'I' is a 3D matrix (RGB) hence, uppercase. 'a' is the 1D audio (time series sequence) and hence lower case. We will update this.
>
> Tuple or set: Apologies, all should be notated as tuples, and k_L and k_S.
>
> **Q3: Dropping points outside the vicinity …, optimal vicinity is not the largest one tested. Why?**
> Again, Great question! Although it might intuitively feel enlarging the vicinity size would help, however please note in the context of the GS learned point based representation, we are typically looking at a scale like ~400k points for a single room scene. Increasing the vicinity would account for adding this huge number of parameters which would make it harder for the field network to be burdened - It is a trade-off. We did try voxelizing the representation, so that we can deal with lesser points (and in turn lesser alpha's) despite increasing the vicinity, but it worsened the performance -we hypothesize this is because the alpha parameter for the points post-voxelization cannot be computed with a simple average (or even a MLP, like anchor points - Scaffold GS [9]). A specialized attention based selection of points for the vicinity might be an interesting future direction to explore, but please note not all points closer to the source or listener are actually (more) helpful.
>
> **Q4: Wouldn't many of the problems in this paper still exist in the monaural listener context as well?**
> It is correct in noting that many of the challenges in this paper are relevant to monaural contexts as well.
> The primary contribution of AV-GS lies in capturing a holistic context of the entire scene and generating context within the 3D scene. Binauralizer arch. is adopted from AV-NeRF (who partly adopted from [1]), not claimed as a novel bit of this work.
> In smaller shoebox rooms (like the one's experimented in Richard et. al ICLR'21, and Office scene in RWAVS), AV-GS's improvement is less pronounced due to limited room geometry impact.
> But, AV-GS shows significant improvements in multi-room more complex scenes (e.g., "house" and "apartment" in RWAVS) with wall occlusions, where AV-NeRF's AV-Mapper is less efficient. (scores in Table 1 of paper)
> Multi-room scenes give rise to multiple wall occlusions - where AV-NeRF's AV-mapper approach is inefficient.
>
> **Q5: Value of T_g**
> Please check W1 above.
>
> **Limitations: Broader impact unclear**
> Thanks, we will clarify. Once an AV-GS has been trained to accurately model room geometry and material properties, synthesizing highly realistic binaural audios in real time should be possible. This could be used to create misleading audio experiences, such as simulating the presence of people or activities that are not actually occurring, which could be used to deceive or manipulate individuals.
> With regards to rendering large scenes, we agree, that generalization to multiple room (4 or 5, or a multi-storey house) is indeed challenging. Additionally, the problem with adopting an explicit points based representation is that the number of points increases drastically for larger scenes, where although capturing fine-grained details is not necessary (as in the case of NVS), a rough geometry is still a crucial requirement for points to be able to guide NVAS.
>
>
> References:
>
> [9] Lu, Tao, et al. "Scaffold-gs" CVPR'24

---

> > ### Comment · Reviewer_hPZr · 2024-08-07
> >
> > Thank you, I think these are helpful clarifications and I appreciate your detailed response. I continue to enthusiastically recommend acceptance.

---

> > > ### Author Response · Authors · 2024-08-08
> > >
> > > We are grateful for the reviewer's encouraging comments. We will incorporate this discussion in the next version.

---

### Official Review · Reviewer_m6xj · 2024-07-12

**Soundness:** 3
**Presentation:** 3
**Contribution:** 3
**Rating:** 6
**Confidence:** 4

**Summary:**

The paper proposes a 3D Gaussian splatting (3DGS) based method for improving the acoustic context modeling for novel view acoustic synthesis (NVAS). The proposed model outperforms multiple existing methods on both simulated and realworld data across different metrics. The method also trains and infers faster than the state-of-the-art AV-Nerf method. The paper also provides model ablations and analyses that help in better understanding the role of different model components.

**Strengths:**

1. The idea of using 3D Gaussian Splatting for better modeling of finegrained geometry and material aware acoustic context for improving the NVAS performance is interesting, novel and works well in practice.

2. The paper reports strong results, where the proposed model outperforms existing methods on both simulated and realworld data

3. The provided model ablations and analyses help in better understanding the contributions of different model components and design choices

**Weaknesses:**

1. Acoustic conditioning with G_a: the paper "averages the context across all points in G_a" (L156-7). Doesn't this lead to loss of finegrained point-specific information? A learned aggregation strategy (for example, by using the attention mechanism) would probably make more sense here?

2. L192-4, "L_v ... non-overlapping":
    i) the insight behind doing this is not clear from the text
    ii) there are no references to support these steps
    iii) there is no ablation for this loss, as well

3. L208, "The audio guidance ... random initialization": won't random initialization during point expansion lead to local 'discontuity'? Did the authors try warm initialization to preserve local 'continuity' by doing nearest neighbor search or bilinear interporation on the existing alphas?

4. Does the model generalize to more challenging scene datasets like Matterport3D [1]?

5. Audio examples: the paper does not seem to provide audio examples and qualitiatively compare their quality with that of the baselines

6. Minor:
   i) L274-5, "4 perspective views ... receiver's position": do these 4 views cover the full 360 degree, otherwise it's not comparable with the field of view of the AVGS model?
   ii) Para in L285-90 looks like repeated text


References:
[1] Matterport3D: Learning from RGB-D Data in Indoor Environments. Chang et al. 3DV 2017.

**Questions:**

Could the rebuttal comment (more) on the following?
1. how the chosen strategy for aggregating the acoustic context in G_a affects model performance. See weakness 1 for details.

2.  the rationale behind using L_m, and how the model performs without the loss. See weakness 2 for details.

3. initialization of alphas during point densification in G_a. See weakness 3 for details.

4. generalization of the model to more challenging datasets like Matterport3D. See weakness 4 for details.

5. my minor comments. See weakness 6 for details.

Also, could the authors anonymously provide a few audio examples from the model and baselines?

**Limitations:**

The paper mentions limitations in Discussion but I could not find any discussion on societal impact.

---

> ### Author Rebuttal · Authors · 2024-08-06
>
> ## Questions (and weaknesses)
>
> **W1, Q1: how the chosen strategy for aggregating the acoustic context in G_a affects model performance.**
> L156-7: "We obtain the condition for binauralization by averaging the context across all points in G_a, post dropping points outside the vicinity of the listener .."
>
> In every iteration a random listener location is sampled.
> Please note from Eq. (3) that the formed context is already conditioned on the direction (position-guidance), which is already an explicit attention.
> The context averaging step happens only for the points within the vicinity (which is a smaller subset of all points that are representing the whole scene, ~400k).
>
> We did try to learn an attention-based weighing among the vicinity points; however we often found that the weight predictor collapses terminating the optimization under a poor local minima (i.e. all points are assigned the same weight, 0.5 for a sigmoidal activated weight predictor) due to the sufficiently varied points provided in every iteration (the vicinity keeps changing drastically across every iteration depending on the sample listener location). One might think to assign weights to all points in the scene instead of  only the ones within vicinity, however this is very compute intensive for a single iteration.
>
> **W2, Q2: the rationale behind using L_v, and how the model performs without the loss.**
> Apologies for not being clear. Our intuition behind introducing alpha is to help us determine each point's contribution to the context used by the binauralizer for generating binaural audio (for a particular listener location). The regularization term encourages lower alpha to prevent over-reliance on a few points, and make the contribution of points (for forming the context) well-distributed, promoting a diverse and representative context which is key for generalization to unseen listener locations.
> [10] proposed a similar regularization term however their purpose is significantly different from ours, where they use it for volume minimization to curb 'visual overlap' and enable faster raymarching.
> We empirically show below that for AV-GS generally lower values of \lambda_a (contribution of regularization loss, Eq. (6)) are preferred.
>
> | lambda_a | MAG (↓) | ENV (↓) |
> |----------|---------|---------|
> | 0.0      | 1.43    | 0.140   |
> | 0.01     | 1.417   | 0.140   |
> | 0.1      | 1.440   | 0.140   |
>
> **W3, Q3: initialization of alphas during point densification in G_a.**
> That's a great question. We tried warm initialization using both repeat (similar to vanilla 3D-GS and Scaffold-GS) as well as the nearest neighbor (NN). In comparison to random initialization, they perform almost the same. We hypothesize this firstly because the number of points (within the vicinity) that satisfy the gradient condition are much smaller, secondly, the densifying step is carried after intervals of 100 iterations, which gives the optimization (which happens every iteration) sufficient space to curb discontinuity, if any. (Not to forget the outlier removal step, carried after an interval of 3k iterations).
>
> | Alpha initialization | MAG (↓) | ENV (↓) |
> |----------------------|---------|---------|
> | random               | 1.417   | 0.140   |
> | repeat               | 1.414   | 0.140   |
> | NN                   | 1.419   | 0.140   |
>
> **W4, Q4: generalization of the model to more challenging datasets like Matterport3D.**
> We currently validate AV-GS on a real-world RWAVS and a synthetic dataset Soundspaces, which is in line with prior work in the field of novel view acoustic synthesis - NAF (NeurIPS'22), INRAS (NeurIPS'22), AV-NeRF (NeurIPS'23). As for challenges, RWAVS (real world dataset) is more challenging than Matterport3D (synthetic), especially the house and apartment scenes involving multiple real rooms in a single scene. Currently both RWAVS and soundspaces-synthetic are publicly available. We can incorporate this into our revised version with similar data (binaural audio and RGB pairs) for Matterport3D.
>
> **W5: Audio examples**
> Because of rebuttal instructions regarding not sharing links, we have provided the AC with an anonymized link to the audio samples. Apologies for the inconvenience. We will also make our code repo and all rendered samples publicly available post the review period.
>
> **W6, Q5: Minor: i) L274-5, do these 4 views cover the full 360 degree? ii) Para in L285-90 - repeated text**
> (i) That is right, all 4 perspective views cover the 360 degrees of listener view. In Table 3, despite the 360 view AV-NeRF falls short of binauralization performance due to lack of a combined holistic scene condition, unlike our method. (ii) Got it, we will remove L285-90 it in the update.
>
> **Limitations:**
> Sorry for the confusion, the impact was provided in the "Appendix / supplemental material". We will update it.
>
> Impact: Once an AV-GS has been trained to accurately model room geometry and material properties, synthesizing highly realistic binaural audios in real time should be possible. This could be used to create misleading audio experiences, such as simulating the presence of people or activities that are not actually occurring, which could be used to deceive or manipulate individuals.
>
> References:
>
> [10] Lombardi, Stephen, et al. "Mixture of volumetric primitives for efficient neural rendering." ACM ToG

---

> ### Comment · Reviewer_m6xj · 2024-08-12
> **Response to rebuttal**
>
> Thanks for the responses. Could you answer/comment on the following?
>
> 1. Lambda_a: what was the value for it in the paper?
>
> 2. Audio examples: thanks for the audio sample, but I think the paper needs a lot more qualitative examples, given that it is an audio spatialization paper. A few suggestions from my end: 1) same mono audio played at different locations in the same scene (2-3 scenes of different sizes and layouts should be evaluated) + qualitative comparison with baselines, 2) repeat 1 for different mono audio samples.
>
> I would urge the authors to add all the results tables and analyses provided in the rebuttal, in the next draft.

---

> > ### Author Response · Authors · 2024-08-13
> >
> > 1. Our current implementation uses lambda_a as 0.01
> > 2. Indeed. We will add all audio samples -multiple scenes and different mono audio source, on our project page.
> >
> > We thank the reviewer for their time and recommending our paper for acceptance. We will incorporate all tables and analyses in the final version.

---

> > > ### Comment · Reviewer_m6xj · 2024-08-13
> > > **Response to rebuttal 2**
> > >
> > > Thanks. I don't have any further questions.

---

### Author Rebuttal · Authors · 2024-08-07

This attached PDF contains the PCA plot and the correlation matrix which is referred in rebuttal response for RHwj - Q2 and DaxK - W2 below.

---

### Decision · Program_Chairs · 2024-09-25

**Decision:**

Accept (poster)

**Comment:**

This is an important and timely contribution. Most technical solutions for novel view acoustic synthesis failed to reliably address both simulated and real datasets while also generating audio that is perceptually reasonable. This paper address all 3 aspects and improves state of the art. The authors back-up the claims with good set of evaluations/ablations. The approach opens up avenues for new material estimation and material swapping methods in audio-visual synthesis and other applications in augmented reality etc.

The key technical novelty comes from showing how to build splatters for audio data, and the overall complexity of the scenes is still limited. Hence, the paper is ready for publication but not at Oral threshold for the conference.

Average rating - 6